# The Effects of Ontario Menu Labelling Regulations on Nutritional Quality of Chain Restaurant Menu Items—Cross-Sectional Examination

**DOI:** 10.3390/nu15183992

**Published:** 2023-09-15

**Authors:** Yahan Yang, Mavra Ahmed, Mary R. L’Abbé

**Affiliations:** Department of Nutritional Sciences, Temerty Faculty of Medicine, University of Toronto, Toronto, ON M5S 1A8, Canada; hailey.yang@mail.utoronto.ca (Y.Y.); mavz.ahmed@utoronto.ca (M.A.)

**Keywords:** menu labelling, restaurant, nutrition policy

## Abstract

Restaurant foods are associated with excessive energy intake and poor nutritional quality. In 2017, the Healthy Menu Choices Act mandated food service establishments with ≥20 outlets in Ontario to display the energy content on menus. To examine the potential impact of menu labelling, nutrition information for 18,760 menu items were collected from 88 regulated and 53 unregulated restaurants. Descriptive statistics were calculated for serving size, energy, saturated fat, sodium and total sugars. Quantile regression was used to determine the differences between regulated and unregulated restaurants. The energy content of menu items from regulated restaurants (median (95% CI): 320 kcal (310, 320)) was significantly lower than those from unregulated restaurants (470 kcal (460, 486), *p* < 0.001). Saturated fat, sodium and total sugars were significantly lower in regulated restaurants (4 g (4, 4), 480 mg (470, 490) and 7 g (6, 7), respectively) than in unregulated restaurants (6 g (6, 6), 830 mg (797, 862) and 8 g (8, 9), respectively, *p* < 0.001). This study showed that menu items from regulated restaurants had smaller serving size, lower levels of energy and nutrients of public health concern compared to those from the unregulated restaurants, suggesting potential downstream beneficial effects of menu labelling in lowering caloric content and nutrients of public health concern in foods.

## 1. Introduction

Although restaurant food consumption is associated with poor diet quality and increased risk of obesity and non-communicable diseases, 54% of Canadians reported eating out at least once a week in 2019 [1,2,3]. The prevalence of take-out/order delivery has also been high and has been further elevated due to the impact of COVID-19 pandemic, which has led to rapid increases in online food delivery services [4,5,6,7]. Interventions aimed at improving diet quality and encouraging healthier behaviors, such as sodium reduction strategies and front-of-pack (FOP) labelling, have to date only targeted prepackaged foods [8,9]. Besides the mandatory trans-fat ban that restricts the use of partially hydrogenated oil in both prepackaged and restaurant foods [8], there are no national nutritional interventions for restaurant foods.

In Canada and globally, voluntary or mandatory nutritional labelling for prepackaged foods has existed for decades [10], however, there are limited regulations in place on the provision of nutrition information for restaurant foods [11]. Studies indicate that consumers struggle to accurately estimate and understand energy and nutrient intake from menu items [12,13]. One strategy to help inform consumers about the nutritional quality of restaurant foods has been the implementation of calorie menu labelling [14]. Since 2008, the U.S. has progressively implemented menu labelling transnationally in chain restaurants with 20 or more outlets, and these restaurants also need to provide complete nutrition information of menu items upon request [15,16,17]. Longitudinal studies suggested a reduction in the energy content of menu items following the implementation of calorie labelling through the elimination of higher-energy items and reformulation to introduce new lower-energy menu items [18,19,20,21,22]. Some studies also found that menu labelling was associated with lower fat and sodium in menu items, a potentially downstream impact of product reformulation [23,24]. Moreover, studies showed that menu labelling is likely associated with lower energy purchased and consumed by consumers, although such impacts may be limited given the small magnitude of changes, and mixed results that have been reported, depending on the study settings and populations [25,26,27].

In 2017, the Healthy Menu Choices Act 2015 was enacted to mandate that food service establishments with 20 or more outlets in Ontario must display the energy content of food items on menus [28]. This impetus was to improve awareness for Ontarians of the energy content of foods and beverages and promote healthier food choices; however, a study assessing the early impact of this legislation in the first year found no reduction in energy content of menu items, and new foods introduced in 2017 were significantly higher in serving size and energy per serving compared with those introduced in 2016 [29]. Since then, there has been no additional assessment of the nutritional quality of restaurant foods. Considering the reformulation of menu items could delay the observable effects, it is unknown whether the effects of calorie labelling on menu items can be seen 4 years post-implementation of menu labelling.

Therefore, the objective of this study was to assess the nutritional quality of menu items in restaurants subjected to menu labelling compared with those that were not, 4 years post the implementation of menu labelling regulation.

## 2. Materials and Methods

### 2.1. Data Collection

Data on Canadian chain restaurant menu items for this study were obtained from the University of Toronto Menu-FLIP (Food Labelling Information Program) database, which was established in 2010 for collecting the nutrition information of Canadian restaurant foods, with detailed information published elsewhere [30]. Briefly, Menu-FLIP for 2020 contains over 20,000 menu items from 141 Canadian chain restaurants, representing over 70% of the market share of food service establishments in Canada. Information collected included identifiers, serving size, energy and 13 core nutrients as listed on the current Nutrition Facts table for prepackaged foods if available [31]. Duplicate menu items of the same size, items with missing or incorrect nutritional information as identified via data validation, toppings/add-ons, atypical offerings, catering and shareable entrées were excluded to increase the accuracy of data and to better represent foods and beverages that would normally be ordered by one consumer. Restaurants were categorized as “regulated” if they have 20 or more outlets in Ontario (i.e., subjected to menu labelling) and “unregulated” if they have less than 20 outlets in Ontario. Food items were categorized into 5 major categories (starters, entrées, sides, desserts, beverages).

### 2.2. Statistical Analyses

Descriptive statistics (mean and median levels with 95% confidence intervals) were calculated by food categories and menu labelling regulatory status for serving size, calories, caloric density and nutrients of public health concern (i.e., saturated fat, sodium and sugar) per serving and per 100 g where data were available. To account for the differences between restaurant venue types, restaurants were characterized as fast-food restaurants (FFR, *n* = 95) if table service was not available and sit-down restaurants (SDR, *n* = 46) if table service was available. Quantile regression was used to calculate the estimated difference in median serving size, calories, caloric density and nutrients of public health concern per serving and per 100 g between regulated and unregulated restaurants, adjusting for restaurant type. Bonferroni corrections were used to correct the family-wise error rate and the significant difference was set at *p* < 0.05. Analyses were performed using R Studio, version 4.0.2 (Boston, MA, USA).

## 3. Results

Of the 141 included restaurants, 88 (62%) had 20 or more outlets in Ontario and were subject to the menu labelling regulations. Overall, menu items from regulated restaurants had smaller serving sizes, lower calories and caloric density than those from unregulated restaurants (*p* < 0.001) (Table 1). By category, entrées from the unregulated restaurants were 121 g larger in median serving size than those from regulated restaurants. Calories per serving overall were significantly lower in the regulated restaurants (−80 kcal (95% CI: −94, −66)) as were calories in desserts (−60 kcal (−84, −36)) and entrées (−151 kcal (−172, −130)). Median caloric density was 66 kcal/100 g lower in regulated restaurants for desserts, 19 kcal/100 g lower for sides, 7.4 kcal/100 g higher for entrées and 16 kcal/100 g lower overall.

Overall, saturated fat, sodium and total sugars per serving and per 100 g were significantly lower in the regulated restaurants in comparison to the unregulated restaurants (*p* < 0.05) (Table 2). By category, median saturated fat per serving was 1 g and 3 g lower in regulated restaurants than in unregulated restaurants for entrées and starters, respectively. Median saturated fat per 100 g was 1.1 g lower in regulated restaurants for starters. Median sodium per serving was 325 mg lower for entrées and 30 mg higher for beverages in regulated restaurants. Median sodium per 100 g was 6 mg higher for beverages in regulated restaurants. Median total sugars per serving was 3 g lower for entrées and 0.8 g higher for sides in regulated restaurants. Median total sugars per 100 g was 0.4 g lower for entrées and 0.7 g higher for sides in regulated restaurants.

## 4. Discussion

This is the first study in Canada to compare the nutritional quality between chain restaurants that were subject to the Ontario menu labelling regulations and those that were not.

Overall, menu items from restaurants subject to menu labelling had smaller serving sizes, lower calories and caloric density, suggesting there have been potential beneficial effects of the legislation on energy reduction in menu items. The median level of energy in entrées was 151 kcal lower in restaurants subject to the menu labelling regulations than in those that were not, representing a reduction equal to nearly 10% of the daily recommendation for energy intake or 23% per meal (1/3 of 2000 kcal) [32]. The lower energy content could be due to reformulation of existing menu items, removal of less healthy items and/or provision of new healthier items. Further research is warranted to fully understand the underlying explanations for these findings. Similar findings have been shown in a U.S. study where menu items from regulated restaurants had lower calorie counts in comparison to those without labelling [19]. It is worth noting that the caloric density for entrées was significantly higher in regulated restaurants than in unregulated restaurants, although the magnitude (7.4 kcal/100 g) was small. Considering the median serving size for entrées was much lower (121 g) in the regulated restaurants, smaller serving size instead of caloric density is likely the main driver of the lower calories.

Additionally, regulated restaurants had significantly lower levels of nutrients of public health concern (saturated fat, sodium and sugar). However, these results should be interpreted with caution since the effect size in certain categories was small and could be of limited nutritional significance and represented less than 10% daily value [33]. Similarly, when standardizing the nutrients of public health concern to 100 g, the differences were negligible (<5% DV), despite being statistically significant. Therefore, there is limited evidence of lower density of nutrients of public health concern in the regulated restaurants, and the differences in nutrient values could mainly be attributed to the smaller serving sizes.

Overall, the lower levels of nutrients of public health concern could be a downstream impact of the regulation, where the lower-energy menu items were also lower in these nutrients, both probably due to lower serving sizes. A U.K. study showed that restaurant foods with voluntary menu labelling had 45% less fat and 60% less salt in comparison to those without labelling, although no serving size data were reported [24]. Besides reducing serving size, a better and healthier way to lower levels of nutrients of public health concern is through product reformulation. Two studies found that removing some salt from foods maintained consumer acceptance, which suggested the feasibility of sodium reduction in restaurant foods [34,35]. Similarly, Patel et al. provided 24 modified menu items by removing certain ingredients (e.g., less sauce) to achieve reductions of up to 210 kcal, 20 g fat, 8 g saturated fat and 1970 mg sodium, and found that these items were acceptable in comparison to items with the original recipes [36]. Therefore, reduction of ingredients and sauces high in nutrients of public health concern shows strong potential for improving the nutritional quality of restaurant foods, yet keeping costs low and taste uncompromised. Since the healthfulness of food is not determined by energy alone, future policies should focus on labelling other nutrients and providing guidelines on improving the overall nutritional quality of restaurant foods. This could be done if governments were to consider requiring restaurant foods to post FOP symbols and warnings on menus when implementing such regulations for prepackaged foods.

Overall, the lower levels of energy and nutrients of public health concern in regulated restaurants suggest potential beneficial downstream impacts of the labelling regulations, where restaurants were incentivized to present and provide healthier menu items to meet the demand of health-conscious consumers. However, our data suggest this was likely achieved through serving size reduction rather than providing foods with a lower density of nutrients of public health concern. Since the regulation requires calories to also be displayed on online menus [28], such an impact could be sustainable given the increased prevalence of online food delivery service platforms during the COVID-19 pandemic [5]. Furthermore, extending the Ontario provincial labelling regulations to the national level, or requiring FOP symbols on restaurant menus, along with other potential interventions such as industry quality standards and population education, could together become complementary and synergistic approaches to improving the diet quality and ultimately the health of Canadians.

### Strengths and Limitations

Although the study data were limited to chain restaurants and were not representative of smaller restaurants, they included larger regional chains that were not subject to the Ontario menu labelling regulations. The database has a large sample size and provides a comprehensive overview of the nutritional quality of menu items in Canadian chain restaurants. The reliability of the nutrition information depends on the restaurants, although under the Food and Drug Regulations they are legally required to publish information that is accurate and up-to-date. We used Atwater calculations and outlier identifications to validate the data and increase accuracy. Since this was a cross-sectional study, we could not determine longitudinal changes in the nutritional quality of menu items. Identifying the mechanism behind the differences in energy and nutrients was beyond the scope of our study. Our analysis was conducted at the item level, and the nature of food types can make comparisons difficult. However, we repeated the analyses with categorical stratifications to better address the variability of menu items. Moreover, there could be other differences between the regulated and unregulated restaurants that were confounders that our analyses were not able to adjust for, although we tried to address a potential confounding factor by adjusting for restaurant types, which showed that the restaurant type was unlikely a source of bias.

## 5. Conclusions

Menu items from restaurants subject to menu labelling regulations were smaller in serving size and lower in energy and nutrients of public health concern than those from unregulated restaurants, suggesting the potential positive effects of menu labelling on the energy and nutritional quality of menu items, although most of the effect might be due to smaller serving sizes. Further longitudinal research can shed light on the mechanistic pathways of effects of the policy, and other policies could focus on interventions targeting serving size and other nutrients of public health concern in addition to energy.

## Figures and Tables

**Table 1 nutrients-15-03992-t001:** Comparison of serving size, calories and caloric density, by category and by restaurant menu labelling regulatory status ^1^.

	Unregulated (*n* ^2^ = 53)			Regulated (*n* ^2^ = 88)				
Category	*n*	Mean (95% CI)	Median (95% CI)	*n*	Mean (95% CI)	Median (95% CI)	Estimate Difference in Median (95% CI)	Bonferroni-Adjusted *p*-Value
Serving Size (g)								
Overall	4138	363 (355, 370)	302 (299, 316)	9065	306 (301, 310)	255 (249, 260)	−34 (−47, −20)	*p* < 0.001 *
Beverages	560	417 (403, 431)	399 (370, 410)	2760	487 (479, 494)	478 (473, 378)	39 (3, 75)	0.2
Desserts	444	139 (131, 147)	123 (115, 130)	890	132 (125, 139)	100 (100, 100)	−15 (−157, 127)	1
Entrées	2372	431 (420, 442)	385 (375, 399)	4476	255 (249, 260)	202 (194, 210)	−121 (−137, −105)	*p* < 0.001 *
Sides	593	217 (201, 234)	150 (144, 166)	824	171 (162, 180)	142 (133, 150)	16 (−2, 34)	0.4
Starters ^3^	169	314 (290, 338)	288 (262, 312)	115	268 (244, 292)	284 (240, 296)	−4 (−34.1, 26.1)	1
Calories (kcal) per serving								
Overall	5912	607 (594, 620)	470 (460, 486)	12,848	424 (418, 430)	320 (310, 320)	−80 (−94, −66)	*p* < 0.001 *
Beverages	680	210 (197, 224)	180 (166, 190)	2975	239 (232, 245)	220 (210, 220)	0 (−15, 15)	1
Desserts	622	381 (362, 401)	340 (320, 360)	1133	333 (318, 348)	260 (257, 280)	−60 (−84, −36)	*p* < 0.001 *
Entrées	3459	772 (755, 790)	660 (640, 680)	7254	527 (518, 535)	420 (410, 432)	−151 (−172, −130)	*p* < 0.001 *
Sides	913	371 (348, 393)	255 (240, 280)	1262	328 (312, 344)	250 (240, 260)	−5 (−29.3, 19.3)	1
Starters	238	732 (679, 785)	624 (590, 740)	224	574 (515, 634)	503 (420, 610)	−115 (−226, −4)	0.2
Caloric density (kcal per 100 g)								
Overall	4138	188 (185, 191)	193 (190, 196)	9065	164 (162, 166)	167 (162, 170)	−16 (−21, −10)	*p* < 0.001 *
Beverages	560	53 (49, 57)	43 (42, 45)	2760	53 (52, 55)	46 (45, 46)	1.6 (−0.5, 3.8)	0.7
Desserts	444	288 (277, 299)	306 (300, 314)	890	259 (251, 267)	241 (229, 247)	−66 (−82, −51)	*p* < 0.001 *
Entrées	2372	196 (193, 198)	199 (196, 202)	4476	204 (202, 206)	210 (209, 213)	7.4 (3.3, 11.4)	0.002 *
Sides	593	200 (192, 208)	195 (188, 200)	824	211 (204, 219)	218 (200, 234)	−19 (−33, −5)	0.04 *
Starters	169	221 (207, 234)	219 (212, 228)	115	190 (169, 211)	200 (132, 232)	−19 (−64, 27)	1

Note: Mean, median with 95% confidence interval of serving size, Calories per serving and caloric density (kcal per 100 g) by category and by restaurant menu labelling regulatory status. Differences in the median of serving size, Calories per serving and caloric density by restaurant menu labelling regulatory status (regulated-unregulated) were calculated using quantile regression, adjusted for restaurant type. * Statistically significance at *p* < 0.05, with Bonferroni adjustment. ^1^ Regulatory Status: Regulated restaurants had 20 or more outlets in Ontario and therefore were subject to the Ontario menu labelling regulations [8], unregulated restaurants had less than 20 outlets in Ontario. ^2^ Number of restaurant chains by restaurant menu labelling regulatory status (regulated/unregulated). ^3^ Since starters were only available in sit-down restaurants, the estimated differences did not adjust for restaurant type.

**Table 2 nutrients-15-03992-t002:** Comparison of saturated fat, sodium and total sugars per serving and per 100 g, by category and by restaurant menu labelling regulatory status ^1^.

	Unregulated Restaurants (*n* ^2^ = 53)			Regulated Restaurants (*n* ^2^ = 88)				
Category	*n*	Mean (95% CI)	Median (95% CI)	*n*	Mean (95% CI)	Median (95% CI)	Estimate Difference in Median (95% CI)	Bonferroni-Adjusted *p*-Value
Per Serving								
Saturated Fat (g)								
overall	5662	10 (9, 10)	6 (6, 6)	10,884	6 (6, 6)	4 (4, 4.3)	−0.9 (−1.1, −0.7)	*p* < 0.001 *
beverages	623	2 (2, 2)	0 (0, 0)	2589	3 (3, 3)	0.1 (0, 0.3)	0 (0, 0)	1
desserts	578	8 (7, 9)	6 (5, 7)	1005	6 (6, 7)	4 (4, 4.5)	−1 (−2.1, 0.1)	0.3
entrées	3330	13 (12, 13)	8 (8, 8.8)	6028	8 (8, 8)	5 (5, 5)	−1 (−1.4, −0.6)	*p* < 0.001 *
sides	887	4 (4, 5)	2.5 (2, 3)	1114	4 (3, 4)	2 (2, 2)	−0.5 (−0.9, −0.1)	0.1
starters ^3^	244	11 (10, 13)	8 (6, 9)	148	8 (7, 10)	5 (4, 6)	−3 (−5.1, −0.9)	0.02 *
Sodium (mg)								
overall	5817	1116 (1089, 1144)	830 (797, 861)	10,908	729 (714, 744)	480 (470, 490)	−170 (−202, −138)	*p* < 0.001 *
beverages	627	210 (197, 224)	40 (35, 55)	2590	129 (123, 134)	90 (85, 95)	30 (17, 43)	*p* < 0.001 *
desserts	637	381 (362, 401)	180 (150, 210)	1004	253 (237, 268)	200 (190, 215)	38 (7, 69)	0.1
entrées	3389	772 (755, 790)	1270 (1240, 1300)	6052	1067 (1046, 1089)	810 (790, 840)	−325 (−369, −281)	*p* < 0.001 *
sides	901	371 (348, 393)	510 (460, 560)	1114	669 (629, 709)	470 (440, 520)	−40 (−103, 23)	1
starters	263	732 (679, 785)	1320 (1150, 1490)	148	1405 (1246, 1564)	1370 (1200, 1460)	40 (−202, 282)	1
Total sugars (g)								
overall	5539	16 (15, 16)	8.1 (8, 9)	10,862	18 (17, 18)	7 (6, 7)	−3 (−3.5, −2.5)	*p* < 0.001 *
beverages	626	32 (30, 35)	31 (29, 33)	2622	40 (39, 42)	36 (35, 37)	1 (−2, 4)	1
desserts	575	29 (27, 31)	22 (21, 23)	1085	30 (28, 31)	20 (19, 21)	−1 (−2.3, 0.3)	0.7
entrées	3267	13 (13, 14)	8 (7.4, 8)	5919	8 (7, 8)	4 (4, 4)	−3 (−3.5, −2.5)	*p* < 0.001 *
sides	812	6 (5, 6)	2 (2.0, 2)	1088	5 (4, 5)	3 (2, 3)	0.8 (0.5, 1.1)	*p* < 0.001 *
starters	259	9 (8, 11)	5 (4, 6)	148	7 (5, 8)	4 (3, 5)	−1 (−2.4, 0.4)	0.8
Per 100 g								
Saturated Fat (g)								
overall	4138	3.1 (3, 3.2)	2.6 (2.4, 2.7)	9065	2.5 (2.4, 2.6)	1.9 (1.8, 2.0)	−0.9 (−1.1, −0.7)	*p* < 0.001 *
beverages	558	0.5 (0.4, 0.6)	0 (0, 0)	2757	0.7 (0.7, 0.8)	0 (0, 0)	0 (0, 0)	1
desserts	446	6.2 (5.7, 6.6)	5.4 (4.1, 6.6)	893	4.4 (4.1, 4.6)	3.5 (3.1, 3.8)	−1 (−2.2, 0.1)	0.4
entrées	2372	3.3 (3.2, 3.3)	3.1 (3, 3.2)	4476	3.2 (3.2, 3.3)	3.1 (3, 3.1)	−0.1 (−0.3, 0)	0.3
sides	593	2.6 (2.3, 2.8)	1.7 (1.5, 1.9)	824	2.3 (2.2, 2.5)	1.4 (1.3, 1.6)	−0.3 (−0.5, 0)	0.4
starters	169	3.3 (2.8, 3.7)	2.8 (2.2, 3.4)	115	2.6 (2.1, 3.1)	1.7 (1.1, 2.1)	−1.1 (−1.8, −0.3)	0.03 *
Sodium (mg)								
overall	4138	333 (325, 340)	333 (327, 342)	9065	333 (325, 340)	269 (260, 278)	−36 (−50, −22)	*p* < 0.001 *
beverages	558	26 (21, 32)	9.9 (8.1, 11.8)	2757	26 (21, 32)	15.5 (14.5, 16.7)	6 (3.4, 8.5)	*p* < 0.001 *
desserts	446	185 (170, 199)	145 (113, 171)	893	185 (170, 199)	130 (121, 153)	−22 (−52, 7)	0.7
entrées	2372	403 (396, 410)	393 (386, 399)	4476	403 (396, 410)	411 (405, 417)	−1.6 (−11.9, 8.7)	1
sides	593	404 (382, 427)	365 (346, 384)	824	404 (382, 427)	401 (380, 432)	−39 (−72, −7)	0.1
starters	169	491 (453, 529)	476 (433, 508)	115	491 (453, 529)	428 (399, 506)	−48 (−111, 15)	0.7
Total sugars (g)								
overall	4138	5.3 (5.1, 5.5)	2.4 (2.3, 2.4)	9065	6.2 (6, 6.3)	2.8 (2.7, 2.9)	−0.2 (−0.4, −0.1)	0.009 *
beverages	558	7.2 (6.6, 7.7)	7.3 (6.5, 8.2)	2757	7.9 (7.6, 8.1)	8.1 (7.8, 8.4)	1.1 (0.1, 2)	0.1
desserts	446	21.3 (20.3, 22.2)	22.3 (21.1, 23.1)	893	23.1 (22.5, 23.8)	22 (21.3, 22.8)	1.6 (0.2, 3)	0.1
entrées	2372	2.8 (2.6, 2.9)	2.1 (2, 2.2)	4476	2.4 (2.3, 2.5)	1.8 (1.8, 1.9)	−0.4 (−0.5, −0.3)	*p* < 0.001 *
sides	593	2.5 (2.1, 2.9)	0.9 (0.7, 1.2)	824	3 (2.7, 3.3)	1.9 (1.6, 2)	0.7 (0.4, 0.9)	*p* < 0.001 *
starters	169	2.6 (2.1, 3)	1.6 (1.2, 2)	115	2.7 (1.9, 3.5)	1.5 (1.3, 1.8)	−0.1 (−0.5, 0.4)	1

Note: Mean, median with 95% confidence interval of saturated fat, sodium and total sugars per serving by category and by restaurant menu labelling regulatory status. Differences in the median of saturated fat, sodium and total sugars per serving and per 100 g by restaurant menu labelling regulatory status (regulated-unregulated) were calculated using quantile regression, adjusted for restaurant type. * Statistically significance at *p* < 0.05, with Bonferroni adjustment. ^1^ Regulatory Status: Regulated restaurants with 20 or more outlets in Ontario and therefore were subject to the Ontario menu labelling regulations, unregulated restaurants had less than 20 outlets in Ontario. ^2^ Number of restaurant chains by restaurant menu labelling regulatory status (regulated/unregulated). ^3^ Since starters were only available in sit-down restaurants, the estimated differences did not adjust for restaurant type.

## Data Availability

The data presented in this study are available for non-commercial research on request from the corresponding author.

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
