# Peer review of "The Effects of Ontario Menu Labelling Regulations on Nutritional Quality of Chain Restaurant Menu Items—Cross-Sectional Examination"

_nutrients, 2023, doi:10.3390/nu15183992_

Round 1

Reviewer 1 Report

The present study investigated the serving size, energy, saturated fat, sodium and total sugars of the 18760 menu items from 88 regulated and 53 unregulated restaurants. The results indicated that the serving size, energy, saturated fat, sodium, and total sugars in regulated restaurants were obviously lower than those of unregulated restaurants.

There are several issues need to be addressed:

1. More than 20000 menu items were collected from 141 Canadian chain restaurants (line 73), but the authors chose 18760 menu items (line 11), please explain it.

2. Is there have any mistake in line 141?

3. Line 116-117, the authors stated that Overall, saturated fat, sodium and total sugars per serving and per 100 g were significantly lower in the regulated restaurants in comparison to the unregulated restaurants (p <0.05) (Table 2), but in the discussion part, when standardizing the nutrients health concern to 100 g, most of the differences were almost negligible (<5% DV) (lin161-162). These two statements are contradiction.

Author Response

Thank you very much for your time in reviewing our manuscript and for the comments. We have done our best to address the comments concerning our manuscript, which are detailed below:

      1. More than 20000 menu items were collected from 141 Canadian chain restaurants (line 73), but the authors chose 18760 menu items (line 11), please explain it.

- Duplicate menu items of the same size, items with missing or incorrect nutritional information as identified via data validation (e.g., results from Atwater calculation did not agree with the reported energy, > 20% difference) were excluded to increase the accuracy of data. Toppings/add-ons, atypical offerings (i.e. limited-time items, salad without dressing), catering and shareable entrées were excluded to better represent foods and beverages that would normally be ordered by one consumer. A total of 21225 items were collected and 2645 items were excluded, leaving the final dataset of 18760 products. We added this information in line 76-80.

     2. Is there have any mistake in line 141?

- This was supposed to be a sub-title in the original submission, it is now removed.

     3. Line 116-117, the authors stated that“Overall, saturated fat, sodium and total sugars per serving and per 100 g were significantly lower in the regulated restaurants in comparison to the unregulated restaurants (p <0.05) (Table 2)”, but in the discussion part,“when standardizing the nutrients health concern to 100 g, most of the differences were almost negligible (<5% DV)” (lin161-162). These two statements are contradiction.

- Thank you for pointing this out. We think although the statistical test showed a significant difference, its nutritional significance can be limited (as discussed in line 161-163) when considering the daily values. To make it more clear, we added 'despite being statistically significant' in line 165.

Reviewer 2 Report

Dear Authors

The topic and results of the study are important from the point of view of the legislator and prove the partial effectiveness of the act. Therefore, I believe that the research should be in-depth and provide a basis for reviewing the regulations.

I propose to slightly change the title, which will be compatible with the purpose of the study:

The effects of Ontario Menu Labelling Regulations on Nutritional Quality of Chain Restaurant Menu Items - Cross-sectional Examination

Author Response

Thank you very much for your review and suggestion. We changed the title as suggested.